# Efficient Autonomous Exploration and Mapping in Unknown Environments

**DOI:** 10.3390/s23104766

**Published:** 2023-05-15

**Authors:** Ao Feng, Yuyang Xie, Yankang Sun, Xuanzhi Wang, Bin Jiang, Jian Xiao

**Affiliations:** 1College of Integrated Circuit Science and Engineering, Nanjing University of Posts and Telecommunications, Nanjing 210023, China; 1221024723@njupt.edu.cn (A.F.);; 2College of Electronic and Optical Engineering & College of Flexible Electronics (Future Technology), Nanjing University of Posts and Telecommunications, Nanjing 210023, China

**Keywords:** autonomous exploration, perception and mapping, deep reinforcement learning, Gaussian process regression, Bayesian optimization

## Abstract

Autonomous exploration and mapping in unknown environments is a critical capability for robots. Existing exploration techniques (e.g., heuristic-based and learning-based methods) do not consider the regional legacy issues, i.e., the great impact of smaller unexplored regions on the whole exploration process, which results in a dramatic reduction in their later exploration efficiency. To this end, this paper proposes a Local-and-Global Strategy (LAGS) algorithm that combines a local exploration strategy with a global perception strategy, which considers and solves the regional legacy issues in the autonomous exploration process to improve exploration efficiency. Additionally, we further integrate Gaussian process regression (GPR), Bayesian optimization (BO) sampling, and deep reinforcement learning (DRL) models to efficiently explore unknown environments while ensuring the robot’s safety. Extensive experiments show that the proposed method could explore unknown environments with shorter paths, higher efficiencies, and stronger adaptability on different unknown maps with different layouts and sizes.

## 1. Introduction

During the autonomous exploration process, robots, without any prior knowledge, rely only on their own sensors and exploration strategies and move through the unknown environment to build a map of the whole circumstance. Autonomous exploration has a wide range of real-life applications, such as the exploration and mapping of unknown environments by rescue robots [1,2], sweeping robots [3,4], and disaster reconnaissance robots [5,6]. In recent years, some remarkable techniques like frontier-based [7,8] and information-based [9,10] methods have emerged. The former method was designed to search for frontier points between the free area and the unknown area and select the optimal frontier point through a global optimization function, which drives robots to the optimal frontier point to build the whole map. The latter utilized Shannon entropy to evaluate the uncertainty of the built map by selecting sensing actions with maximized mutual information (MI), which drives the robot to explore unknown areas. However, due to the complexity and uncertainty of the unknown environment, it is very challenging to formulate a precise and general global optimization function [11]. Therefore, most methods utilize a greedy strategy that minimizes the distance traveled, maximizes the information gained, or both, with fixed weights as the global optimization function without considering future planning. This may lead to a reduction in the efficiency of the robot’s exploration and an increase in the total length of exploration paths [12]. In addition, with the richness of map information, each decision process can take an extended amount of time [13], and Deep Reinforcement Learning (DRL) based approaches are considered to solve these problems.

Since DRL gained great success in video games such as Atari [14], some researchers have begun investigating DRL’s application in autonomous exploration. Currently, there are two main categories of DRL-based exploration methods. One utilizes raw sensor data (such as depth images [15,16,17] and LiDAR data [18,19]) as input. This approach is usually an end-to-end approach that drives robot exploration by establishing mapping relationships between raw sensor data and robot control variables. It usually requires consideration of avoiding obstacles in an unknown environment when training, which undoubtedly makes the system more difficult to train [11]. Another type of approach uses environmental features (local environmental features [20,21,22] or global environmental features [23,24,25,26]) as input. This method usually predicts the robot’s next direction of motion or target point directly, without considering navigation and obstacle avoidance. Of these, the former uses local features as input and trains robots that are robust to any layout and environment size. However, due to the lack of global information, it often falls into the local optimum problem [27]. The latter uses global features as input which usually has a huge state space, creating a great challenge for the efficient convergence of DRL. As a result, researchers generally supplement the training of the global environment feature method with environment topology [25] and boundary locations [28], but this lowers the adaptability of the system in various environments [12].

Moreover, autonomous exploration methods for unknown environments usually maintain high exploration efficiency in the former stages of exploration while suffering from fairly low exploration efficiency in the later stages of exploration [11]. The main reason for this is the regional legacy issues. Throughout the exploration process, autonomous exploration methods usually pursue unexplored regions with higher gain according to their exploration strategies, resulting in some unexplored regions with less gain being left in various corners of the map. In the latter stages of exploration, the robot takes a long path cost to complete the final exploration. Existing exploration methods usually do not actively address regional legacy issues. Instead, by passively setting an exploration threshold, these methods stop exploration when the exploration ratio reaches the exploration threshold in order to maintain high exploration efficiency [8,29]. However, this directly affects the quality and integrity of the built maps.

To address the common problems of existing exploration methods, such as the region legacy issues and the challenge of balancing the optimal exploration path and environmental robustness with a single exploration strategy, we propose a Local-and-Global Strategy (LAGS) algorithm that integrates a local exploration strategy and a global perception strategy. The local exploration strategy emphasizes high-gain regions in the local map to expedite exploration of unknown areas while ensuring that the algorithm is robust in the environment; the global perception strategy focuses more on maximizing future rewards, guiding the robot to choose the optimal exploration path through the perceived global information to avoid inefficiencies in the later stages of exploration. The LAGS algorithm can proactively solve the regional legacy issues in the autonomous exploration process, increasing the robot’s exploration efficiency. In time-sensitive tasks such as human search and rescue and disaster reconnaissance, the LAGS algorithm allows faster exploration of unknown environments to speed up the search for victims. Our contributions are listed below:We analyze the impact of the regional legacy issues on the efficiency of exploration and propose a LAGS algorithm that can solve the regional legacy issues during the exploration process and improve the efficiency of exploration.We combine a local exploration strategy with a global perception strategy, solving the problem that a single exploration strategy is challenging to balance between optimal exploration paths and environmental robustness.We use Gaussian process regression (GPR) and Bayesian optimization (BO) sampling points as candidate action points for the robots. Compared to the classical frontier-based candidate point selection methods, our approach ensures that each candidate action point is safe and has a higher MI gain.Extensive experimental results obtained on various maps with different layouts and sizes show that the proposed method has shorter paths and higher exploration efficiency than other heuristic-based or learning-based methods.

The paper is organized as follows: Section 2 presents related work on the development of autonomous exploration in heuristic-based and learning-based terms; Section 3 describes the system pipeline of our proposed method and the problem definition for the autonomous exploration task; Section 4 describes the GPR prediction of MI reward surfaces and the process of determining candidate action points by BO; Section 5 presents and analyzes the proposed algorithm and network structure; Section 6 conducts a series of simulations and scalability experiments and compares the performance between the different methods; Section 7 summarizes the existing work and discusses possible future research directions.

## 2. Related Works

The development of autonomous exploration can generally be divided into two categories: the heuristic-based method and the learning-based method. The heuristic-based method is still the most popular method in solving autonomous exploration problems [24], while the learning-based method is considered to be the most promising method and is currently a hot research topic [30].

Among all the heuristic-based methods, frontier-based and information-based algorithms are the most classical. The frontier algorithm was first proposed by Yamauchi [31], who utilized a depth-first search algorithm to drive a robot to the nearest frontier point to complete the exploration of the map. Researchers have proposed various improvements to this method, such as using a rapidly exploring random tree (RRT) algorithm to identify candidate frontier points [32] and also formulating a reasonable and efficient global optimization function to select the optimal frontier point [33,34]. Since the safety of the target points selected based on the frontier algorithm is uncertain, Selin et al. [35] combined the Frontier Exploration Planning (FEP) and Next Best View Planning (NBVP) algorithms to enable the robot to explore more unknown areas as possible while ensuring its safety. In the information-based methods, Bourgault et al. [36] introduced Shannon entropy to evaluate the uncertainty of constructing maps during exploration. Based on this, Ila et al. [37] defined an MI gain and selected paths that minimize map entropy to explore the environment. Julian et al. [38] improved the representation and calculation of MI by proposing a method to calculate MI using a beam-based model. Due to the high cost of calculating MI, Bai et al. [39,40] proposed a method to predict MI based on GPR, where the MI results were only calculated from the training samples of a two-dimensional Sobel sequence. However, this method can only give an approximate prediction of MI. As a result, the BO-based active sampling method was proposed later to improve the prediction accuracy of MI. In general, the results of heuristic-based methods mainly depend on the formulated global optimization function and current observations of the environment without considering future exploration. As a result, longer exploration paths and lower exploration efficiency can always be seen in most heuristic-based methods modeling processes [30].

With the rapid development of deep learning (DL) and reinforcement learning (RL), learning-based methods are considered an effective approach for solving the problems above. DRL, merging DL and RL, has been widely used as an effective sequential decision-making tool in areas such as unmanned aerial vehicle (UAV) navigation [41], multi-objective path planning [42], and computer games [43]. Hence, researchers have considered defining autonomous exploration tasks as a sequential decision problem and using DRL to solve it. Juliá et al. [44] modeled autonomous exploration as a partial observation Markov decision process (POMDP) and used direct policy search to solve it. Tai and Liu [45] used a convolutional neural network (CNN) to extract input image features and trained a deep Q-network (DQN) based control network. However, this algorithm only achieved collision-free roaming in unknown environments. Bai et al. [27] constructed a supervised learning problem to predict the next direction of movement by feeding local features of an environmental map into a neural network. However, this algorithm can only perceive local information about the map and often falls into local solutions. Thus, Chen et al. [12] trained a recurrent neural network (RNN) and added a Frontier Rescue (FR) strategy to solve the local solution, which still suffers from path redundancy during the exploration process. Dooraki and Lee [46] trained decision networks using deep Q-learning neural networks and deep Actor-Critic neural networks with a memory module, respectively, to achieve autonomous exploration in simple environments. However, the networks struggled to achieve effective convergence due to their end-to-end exploration approach. Therefore, Li et al. [11] proposed a generic exploration framework containing decision, planning, and mapping modules by decomposing the exploration process and using an edge segmentation task in the decision module to speed up the convergence of the neural network. Nevertheless, the action points of this algorithm were obtained by rasterized sampling on the occupancy map and thus possessed a large action space, which undoubtedly increased the computational cost of the system. Zhu et al. [12] used DRL to predict the order of visits to unexplored subregions and used the NBV selection algorithm to find optimal target points, yet this approach can only be utilized in regular structures such as office buildings. Niroui et al. [28] combined DRL with a frontier-based method by using the Asynchronous Advantage Actor-Critic (A3C) algorithm and frontier point locations for training, enhancing the robot’s adaptability to unknown environments. However, this algorithm is not stable enough and often suffers from inefficient exploration in the latter stages of exploration.

Among all the effective works, we note that few researchers consider the regional legacy issues during exploration. A common way is to stop exploration in time when the exploration ratio reaches a certain threshold to avoid inefficiencies later in the exploration process. For example, in recent work, Zagradjanin et al. [8] and Li et al. [11] used the exploration ratio to define a hyperparameter for balancing the path length and exploration efficiency during exploration. As far as we know, we are the first team trying to address the regional legacy issues in the exploration process.

## 3. System Overview and Problem Formulation

### 3.1. System Overview

Figure 1 shows the complete system pipeline. The occupancy map with the robot position is used as the input to the system, and the candidate action points are determined by BO sampling [47,48]. The candidate action points are calculated with the occupancy map, robot position, and candidate action points according to the local exploration strategy and the global perception strategy. The local exploration strategy considers the local occupancy map and the candidate action points as its input, and the global perception strategy chooses the global occupancy map and the candidate action points as its input. The evaluated candidate action points are weighted together, and the highest-scoring candidate action point is selected as the target point for the next movement. The current position of the robot and the target point are sent to the navigation module, which plans the optimal path and drives the robot to the target point for exploration.

### 3.2. Problem Formulation

The goal of autonomous exploration by a robot can be defined as finding an optimal exploration path $L^{*}$ for the construction of the whole unknown map [12]. We decompose the optimal exploration path $L^{*}$ into a set of target points $F=\left\{X_{1},X_{2},\ldots ,X_{T}\right\}$ so that the goal of autonomous exploration can be converted into finding a set of optimal points $F^{*}$. The robot can complete the construction of the whole unknown map by traversing the target points in the point set in turn. The best points set $F^{*}$ can be expressed as follows:(1)$$F^{*}=\underset{X_{1:T}}{argmin}\left\{H(m_{t})+\sum_{t=1}^{T}L(X_{t}|m_{t})\right\}$$
where $H(m_{t})$ is the Shannon entropy of the currently occupied map $m_{t}$, T is the final time step, and L is the length between $X_{t}$ and $X_{t-1}$ on the path.

In fact, the optimal points finding process in autonomous exploration can be seen as a sequential decision process, where the robot chooses a series of actions to maximize the accumulation of future rewards [14]. Markov decision process (MDP) [20,26] is a commonly used decision framework used in solving the sequential decision problems mentioned above and can be formulated as a tuple <S,A,T,G,γ>. S is a finite set of states, and in the proposed system, s∈S is the information currently known by the robot, including the current position, current observations of the environment, and candidate action points on the current environment. A is the set of actions that the agent can perform. $T(s^{\prime }|s,a)$ gives the distribution of transformations over the current state s and the set of actions a to the next state $s^{\prime }$. G is the future discounted reward, which is denoted as follows:(2)$$G_{t}=r_{t}+\gamma r_{t+1}+\gamma^{2}r_{t+2}+\cdots {+\gamma }^{T-t}r_{T}=\sum_{t^{\prime }=t}^{T}\gamma^{t^{\prime }-t}r_{t^{\prime }}$$
where $G_{t}$ refers to the expected reward between the time t and the final time step T, and γ is the discount factor.

## 4. Bayesian Optimization Based Sampling

This section describes how the process of determining a robot’s candidate action points through BO sampling is performed. The overall process is shown in Figure 2. Firstly, $N_{GPR}$ sampling points from a 2D Sobel sequence [49] are selected to form the initial training sample. GPR [50,51] predicts the posterior mean and variance of the currently occupied map based on the initial training sample. The posterior mean represents the predicted MI reward surface of the current occupancy map, and the variance represents the degree of uncertainty of the current predicted MI reward surface. After GPR convergence or the maximum number of iterations $N_{BO\_max}$ is reached, we use the posterior mean as the MI reward surface of the currently occupied map and select $N_{G\_action}$ sampled points with the highest MI gain as the candidate action points for the robot. To prevent the robot from having no candidate action points in the local environment, we also sample the local map $N_{L\_BO}$ times (BO sampling) and select $N_{L\_action}$ sample points with the highest MI gain, which are added to the candidate action points of the robot. After several experiments, we set $N_{GPR}$ to 40, $N_{BO\_max}$ to 60, $N_{G\_action}$ to 20, $N_{L\_BO}$ to 20 and $N_{L\_action}$ to 5.

### 4.1. Mutual Information Gain

In the autonomous exploration process, the robot needs to consider a combination of map and positional uncertainties to select a preferable candidate action point. Since the environment is unknown (because we cannot directly know the uncertainty of the currently built map), we use the Shannon entropy [52,53] as the uncertainty of the currently built map instead. The Shannon entropy is defined on the occupancy grid map m as follows:(3)$$H(m)=-\underset{i}{\sum }\underset{j}{\sum }p\left(m_{i,j}\right)\log p\left(m_{i,j}\right)$$
where $p(m_{i,j})$ represents the occupancy probability of the cross grid of row *i* and column *j*. Each grid has three occupancy probabilities: idle, occupied, and unknown. Idle and unknown cells do not contribute entropy, and the probability of unknown cells is defined as $p(m_{i,j})=0.5$. Each unknown cell contributes one unit of entropy.

MI represents the extent to which the current measurement in the occupancy map reduces entropy, i.e., the reduced uncertainty in the robot position associated with all cells [38]. The MI reward function is defined as follows:(4)$$I\left(m_{t},x_{t+1}\right)=H(m_{t})-H\left(m_{t}|x_{t+1}\right)$$
where $H(m_{t})$ is the entropy of the currently occupied map $m_{t}$ and $H(m_{t}|x_{t+1})$ denotes the expected entropy of a new map constructed based on sensor observations after selecting action $x_{t+1}$ on the occupied map $m_{t}$. Directly taking action with the maximum MI gain can not guarantee an optimal exploration path [27]. Therefore, we choose several action points with high MI gain as candidate action points and submit them to DRL to select the optimal candidate action points to perform to guarantee the optimal exploration path.

### 4.2. Gaussian Process Regression

Since the direct computation of MI reward surfaces is costly, we use GPR to predict MI reward surfaces as a way to reduce the computational cost of the system. We assume a set of training data $x=\left\{x_{1},x_{2},\ldots ,x_{n}\right\}$, where the corresponding mutual information $I(m,x_{t})$ for all $x_{t}\in x$ has been computed and forms the output set $y=\{y_{1},y_{2},\ldots ,y_{n}\}$. The GPR is constructed using the training data x and its corresponding output y, which is used to predict the MI gain at the specified location of the currently occupied map. The GPR will estimate the output value $y_{*}$ and the corresponding covariance $cov(y_{*})$ associated with the test data $x_{*}$ by the following equation:(5)$$y_{*}=k\left(x_{*},x\right){\left[k(x,x)+{\sigma_{n}}^{2}I\right]}^{-1}y$$
(6)$$\mathrm{cov}\left(y_{*}\right)=k\left(x_{*},x_{*}\right)-k\left(x_{*},x\right){\left[k\left(x,x\right)+\sigma_{n}^{2}I\right]}^{-1}k\left(x,x_{*}\right)$$

In the above equation, $y_{*}$ is the valuation $I(m,x_{*})$ of the test data $x_{*}$, $cov(y_{*})$ is the covariance associated with the output $y_{*}$, ${\sigma_{n}}^{2}$ is the vector of Gaussian noise variances associated with the training output y, and $k(x,x^{\prime })$ is the kernel function (also known as the covariance matrix). The commonly used Matérn kernel is utilized to describe the correlation between the input data x and $x^{\prime }$ [50], which is defined as follows:(7)$$k\left(x,x^{\prime }\right)=\frac{2^{1-v}}{\Gamma (v)}{\left(\frac{\sqrt{2v}\left|x-x^{\prime }\right|}{\ell }\right)}^{v}K_{v}\left(\frac{\sqrt{2v}\left|x-x^{\prime }\right|}{\ell }\right)$$
where v is a parameter used to vary the covariance smoothness, ℓ is the feature length, Γ is the gamma function, and $K_{v}$ is the modified Bessel function. Compared to another commonly used radial basis function (RBF) kernel, the Matérn kernel can successfully simulate sharp changes in the mutual information reward surface due to the presence of an obstacle.

### 4.3. Bayesian Optimal Sampling

The candidate sampling actions suggested by BO are calculated using the acquisition function [47]. There are two categories of optimization for the acquisition function: exploration, where selection is made in unexplored regions with high uncertainty, and exploitation, where selection is made in the vicinity of the existing maximum. Any collection function used must be traded off between exploration and exploitation. We use the Gaussian Process Upper Confidence Bound (GP-UCB) algorithm [50] as our collection function, the GP-UCB is expressed as follows:(8)$${x_{sample}}^{*}=\underset{x\in C_{action}}{argmax}\mu (x)+\kappa \sigma (x)$$
where κ is the trade-off parameter between exploration and exploitation, $C_{action}$ is the action point with the exact resolution as the occupied map. µ(x) and σ(x) are the mean and variance predicted by the GPR, and ${x_{sample}}^{*}$ denotes the selected optimal sampling point. In the proposed system, we prefer distributing the robot candidate action points evenly across multiple high-value regions rather than concentrating on a single high-value region. Therefore, we configure the acquisition function to favor exploration over exploitation as much as possible.

Figure 3 illustrates how to determine candidate action points for the robot using BO sampling. Figure 3a shows the currently built occupancy map, robot position, and local map. Figure 3b shows the “ground truth” of MI, which is the actual MI reward surface for the current occupancy map. Figure 3c shows the initial training sample for GPR and the predicted MI reward surface based on this. Figure 3d shows the MI reward surface and training samples after 60 BO iterations. Figure 3e shows the MI reward surface and the local sampling points predicted based on the local map. Figure 3f shows the selected local and global candidate action points.

## 5. DRL-Based Decision Method

In this paper, we use a direct strategy to calculate the unexplored regions with the highest gain in local exploration. Meanwhile, global perception trained using DRL guides local exploration by extracting global information to plan the optimal exploration path.

### 5.1. Local Exploration and Global Perception

#### 5.1.1. Local Exploration

The objective of local exploration is to find the candidate action point ${x_{local}}^{*}$ with the highest MI gain in the current local region to explore the unknown region as fast as possible, and its objective function is expressed as follows:(9)$${x_{local}}^{*}=\underset{x_{t}\in L_{action}}{argmax}I\left(m_{local},x_{t}\right)$$
where $m_{local}$ represents the current local occupancy map, $L_{action}$ represents the set of candidate action points on the current local map, and ${x_{local}}^{*}$ represents the selected local optimal candidate action points. We define a heuristic reward score for the local exploration strategy and normalize the score: score∈[−1,1]. After empirical testing, we set 70% of the maximum possible MI gain as the upper bound of the score $I_{ht}(m_{local},x)$, as well as normalized the MI gain obtained by the robot after selecting a candidate action point. When the robot is too close to an obstacle after selecting a candidate action point or the MI gain obtained is less than the threshold $I_{lt}(m_{local},x)$ (this threshold is slightly greater than zero), it will be given a penalty of −1. This provides adequate safety for the robot and prevents it from repeatedly exploring the explored regions. The reward score can be expressed as:(10)$${score}_{local}=\left\{\begin{array}{l} 1\;\;\;\;\;\;\;I\left(m_{local},x_{t}\right)\geq I_{ht}\left(m_{local},x\right)\\ \frac{I(m_{local},x_{t})}{I_{ht}(m_{local},x)}\;\;\;I_{lt}(m_{local},x)<I(m_{local},x_{t})<I_{ht}(m_{local},x)\\ -1\;\;\;\;\;\;I\left(m_{local},x_{t}\right)\leq I_{lt}\left(m_{local},x\right)\;or\;close\;to\;the\;obstacle \end{array}\right.,$$
where ${score}_{local}$ is the calculated score value from the local candidate action points and $I(m_{local},x_{t})$ denotes the MI gain obtained by selecting the candidate action point $x_{t}$ on the current local occupancy map $m_{local}$.

#### 5.1.2. Global Perception

Global perception focuses on small unexplored regions that are closer to the robot. When there are two unexplored regions with similar distances and different gains, global perception prefers to select the unexplored region with the smaller gain to compensate for the regional legacy issues caused by excessive greed in local exploration. Its objective function can be expressed as follows:(11)$${x_{global}}^{*}=\underset{x_{t}\in G_{action}}{argmax}\left\{-\phi \frac{I\left(m_{global},x_{t}\right)}{I_{ht}\left(m_{global},x\right)}-\varphi \frac{L(x_{t})}{L_{max}}\right\}$$
where $G_{action}$ denotes the set of candidate action points on the current global occupancy map, ϕ and φ are weight coefficients, and ϕ<φ, ${x_{global}}^{*}$ denotes the selected global optimal candidate action point. L(·) is the A* path length between the robot and this candidate action point. Global perception also has an important function. When local exploration falls into a local optimal solution, global perception will guide the robot to the unexplored region to continue exploration. In addition, global perception is equipped with a terminal action that will decide whether to take a terminal action to stop exploration when the exploration area is determined to be larger than a threshold, the definition of a terminal action and the reward received for taking that action are shown as follows:(12)$$r_{t}=\left\{\begin{array}{l} 1\;\;\;\;\;if\;the\;ratio\;of\;explored\;region\;in\;map\;\rho >0.95\\ -0.2\;\;otherwise \end{array}\right.$$

In the above equation, ρ is a hyperparameter that indicates the proportion of the region to be explored. During the empirical evaluation, we set it to 0.95.

### 5.2. Network Structure

A combined image is constructed for the global perception network inputs, including the current occupancy map, robot position, and the candidate action points on the current occupancy map. The robot position and candidate action points are each plotted on a blank map of the same size as the occupancy map. The aim is to enable the network to take full advantage of all the critical information useful for decision-making.

The network structure is shown in Figure 4. Since the superior performance of feature extracting, we use four convolutional layers to extract features from the input image. After each convolutional operation, we use a rectified linear unit (RELU) to remove the negative feature values. The features extracted by the convolutional layers are classified by a fully connected layer and transferred to a long short-term memory (LSTM) unit to ensure that the network considers previous critical information when making decisions. The output of the LSTM unit is used directly by the Actor and Critic layers. The output of the Actor layer is passed through the Sigmoid activation function to obtain a weight parameter ω∈(0,1). The Filter layer uses the weight parameter ω in the evaluation function to evaluate the score value of each input candidate action point. The evaluation function consists of the A* path length between the robot and the candidate action point and the MI gain of the candidate action point [28]. The evaluation function is formulated as follows:(13)$${score}_{global}=\omega \times \overset{-}{g}+(1-\omega )(1-\overset{-}{d})$$
where ${score}_{local}$ is the calculated score value from the global candidate action points, $\overset{-}{d}$ and $\overset{-}{g}$ are the normalized distance and MI gain. The score values of candidate action points evaluated by both the local exploration strategy and the global perception strategy are weighted and superimposed by the score values of both. The superimposition equation is shown below:(14)$${score}_{finsh}={score}_{local}+\eta {score}_{global}$$
where ${score}_{local}$ and ${score}_{global}$ denote the scores evaluated by the local exploration strategy and the global perception strategy, respectively, and η is the scale factor. Once the scores of all candidate action points are determined, the robot will select the candidate action point with the highest score for the next exploration.

### 5.3. Asynchronous Advantage Actor–Critic Algorithm

We train our network with the A3C algorithm [54,55] based on the Actor–Critic framework. It combines value-based and policy-based methods and has two outputs, one corresponding to the policy π and another output of the state value V. The loss function of the A3C algorithm typically consists of three terms [56]: the policy gradient loss, value residual, and strategy entropy regularization, denoted respectively as follows:(15)$${Policy}_{loss}=-\underset{\tau ∼\pi_{\theta^{\prime }}}{E}\sum_{t=0}^{T}{\log\pi }_{\theta \prime }\left(a_{t}|s_{t}\right)A\left(s_{t},a_{t};\theta ,\theta_{v}\right)$$
(16)$${Value}_{loss}=\underset{\tau ∼\pi_{\theta }}{E}\left[\frac{1}{2}\sum_{t=0}^{T}{\left(R_{t}-V\left(s_{t};\theta_{v}\right)\right)}^{2}\right]$$
(17)$${Entropy}_{loss}=-H\left(\pi_{\theta^{\prime }}\left(a_{t}|s_{t}\right)\right)$$
where the strategy entropy regularization term ensures the diversity of actions and enhances the robot’s ability to explore the environment. The total loss function is shown as follows:(18)$${Total}_{loss}={Policy}_{loss}+\alpha *{Value}_{loss}+\beta *{Entropy}_{loss}$$

In the above equations, H is the entropy of the strategy, α is the weight coefficient of the value residual term, β is the weight coefficient of the strategy entropy regularization term, and θ and $\theta_{v}$ are the parameters of the strategy and value functions, respectively. $A(s_{t},a_{t};{\theta ,\theta }_{v})$ is an estimate of the dominance function, indicating the dominance of the chosen action $a_{t}$ relative to the average in state $s_{t}$, which can be expressed as follows:(19)$$A\left(s_{t},a_{t};\theta ,\theta_{v}\right)=R_{t}-V\left(s_{t};\theta_{v}\right)$$
(20)$$R_{t}=G_{t}+\gamma^{k}V\left(s_{t+k};\theta_{v}\right)$$
where $G_{t}$ and γ have the same meaning as in Equation (3), k is the total steps from the current step t to the final step T, and V is an estimated function of the additional benefit gained by taking action in a certain state.

The A3C algorithm can start multiple threads (workers), each of which synchronizes the latest network parameters from A3C’s global network and performs independent exploration, training, and learning [56]. When this exploration ends or reaches the maximum step size T, the gradient accumulated by itself is updated to A3C’s global network. Then, the latest network parameters are synchronized and repeated until convergence. θ and $\theta_{v}$ are updated as follows:(21)$$d\theta =d\theta +\nabla_{\theta^{\prime }}\log\pi \left(a_{t}|s_{t};\theta^{\prime }\right)\left(R_{t}-V\left(s_{t};\theta_{v}\right)\right)+\beta \nabla_{\theta^{\prime }}H\left(\pi \left(a_{t}|s_{t};\theta^{\prime }\right)\right),$$
(22)$$\theta_{v}=d\theta_{v}+\partial {\left(R_{t}-V\left(s_{t};\theta_{v}\right)\right)}^{2}/\partial \theta_{v},$$
where H is the entropy of the strategy and β is the weighting factor of the strategy entropy regularization term.

## 6. Experiments

In this section, we train our algorithm in a simulation environment, then compare it with different algorithms to evaluate our algorithm’s performance and analyze its generalizability on maps of varying layouts and sizes. Finally, we summarize the experimental results.

### 6.1. Evaluation Indicators

To analyze the performance of our algorithm, we use the explored region rate, average path length, and exploration efficiency as evaluation metrics, which are expressed as follows:
Explored region rate ζ: This metric evaluates the completeness of the map built by the robot during exploration, and it is defined as
(23)$$\zeta =\frac{N_{f\_explored}}{N_{f\_real}}$$where $N_{f\_explored}$ is the number of free spaces explored and $N_{f\_real}$ is the number of free spaces in the actual map;
2.Average path length $\overset{-}{L}$: This metric evaluates the average path length taken by the robot in the total set of trials, and it is defined as
(24)$$\overset{-}{L}=\frac{\sum L_{i}}{N_{episodes}}$$
where $N_{episodes}$ is the number of trials, and $L_{i}$ refers to the length of the path moved on the ith trial;3.Exploration efficiency Κ: This metric evaluates the entropy reduction of the robot after moving a unit distance on average over the total set of trials, and it is defined as
(25)$$Κ=\frac{\sum \left(H(m_{T_{i}})-H(m_{0})\right)}{\sum L_{i}}$$where H(m) is the entropy of the occupied map m, and $T_{i}$ and $L_{i}$ refer to the number of steps moved and the length of the path in the ith trial, respectively.

### 6.2. Simulation Setup

We design a Python-based simulation platform to train the intelligent robot, using a beam-based sensor model to simulate the robot’s exploration of its environment and an A* algorithm to plan the robot’s path to a target point. We assume that the robot can move omnidirectionally and has an all-round 360° field of view with a resolution of 0.5° and that the sensor can detect a circular area with a radius of 20 pixels. We train and test our algorithm on maps of different layouts and sizes selected from HouseExpo (a sizeable indoor layout dataset) constructed in [57]. Figure 5 and Table 1 show the details of the maps.

### 6.3. Training Setup

We train our algorithms on an Intel (R) Core (TM) i9-9900K CPU at 3.60 GHz without using a GPU for acceleration. We train three agents simultaneously to interact with the environment in training maps 1, 2, and 3, respectively, with the robot’s initial position being randomized for each exploration. After several experiments, the hyperparameters of the A3C algorithm and network structure used for tests are shown in Table 2 and Table 3, respectively.

### 6.4. Algorithm Comparison

We use several heuristic-based and learning-based methods as baselines to evaluate the performance of our method, formulated as below:NF (nearest frontier). The method is explored by selecting the nearest frontier point to the robot as the target point.MI (maximum mutual information). This method calculates the MI gain for each action point and explores it by selecting the action point with the maximum MI gain.LS (local exploration strategy). We modified the source code provided by Chen et al. [13] and applied it to our environment. The action points of the method consist of 40 Sobel sampling points in the vicinity of the robot, and the goal is to find and execute the action point with the highest gain in the local area. When the LS reaches a “dead end”, it is guided to the nearest frontier point using the FR strategy.GS (global exploration strategy). We replicate the method proposed by Niroui et al. [28] in our setting. The method uses the frontier points of the currently occupied map as action points, with the goal of maximizing the total information gained from the robot’s exploration path.

In these methods, NF and MI are heuristic-based methods, and LS and GS are learning-based methods. Heuristic-based methods explore unknown environments based on global optimization functions, and learning-based methods explore unknown environments based on autonomously learned strategies. We first tested on Test Map 1, shown in Figure 5, which was the same size as the training map but had a different layout. All methods were tested 30 times with a random initial location, and each exploration does not stop until the environment is fully explored. The combined performance is shown in Figure 6. The curves represent the proportion of explorations at the average path length.

As can be seen in Figure 6, the LAGS algorithm had better performance and shorter paths than other algorithms. The average path length of LAGS was 18.8%, 49.3%, 26.4%, and 38.4% shorter than the average path length of NF, MI, LS, and GS, respectively. We can see that MI, LS, and GS showed good exploration efficiency in the early exploration stage. However, the exploration efficiency decreased to varying degrees in the latter stage of exploration due to the influence of the regional legacy issues, and the decrease was especially obvious for MI and GS. LAGS and NF were less affected by the regional legacy issues. They maintained a more stable exploration efficiency throughout the exploration process, with the overall exploration efficiency of LAGS being significantly higher than that of NF.

In order to better analyze the exploration performance of LAGS and other algorithms, we compared the exploration paths of these algorithms at the initial positions (10,10), (70,35), and (35,50), respectively. The comparison results are shown in Figure 7. Compared with other algorithms, LAGS could find more reasonable exploration paths at different initial positions, and there were almost no duplicate explorations and redundant paths. Due to MI’s one-sided exploration strategy for high gain and LS’s lack of global information, there was always redundancy in their exploration paths. NF depends on the map’s features due to its non-learning exploration strategy, which makes it exhibit different exploration performances at different initial locations. However, GS could plan a reasonable exploration path in some cases, but it uses more path cost to complete the final exploration in the later stages of exploration.

LAGS can plan reasonable exploration paths, mainly because it considers and solves the regional legacy issues in the exploration process. We visualize the exploration process of the LAGS algorithm at an initial position of (10,10), which is depicted in Figure 8. Early in the exploration, the robot selects target points with high MI gain for exploration to explore the environment faster. In steps four and nine, the robot has several selectable unexplored regions. Global perception guides the robot to select unexplored regions that are close together and smaller in size to avoid regional legacy issues later in exploration. After step 12, the LAGS takes a terminal action to stop exploration.

### 6.5. Summary Analysis

To analyze the generalizability of LAGS, we tested it in test maps 1, 2, and 3 with different layouts and sizes, as shown in Figure 5. We tested each test map 50 times with a random initial position. The algorithm chooses at its discretion whether to take a terminal action to stop exploration during the exploration process. The test results are shown in Table 4 and Figure 9.

As can be seen from Table 4 and Figure 9, LAGS has the shortest exploration path and highest exploration efficiency in all the tested maps. NF and MI have higher exploration ratios because the heuristic methods keep exploring the environment until there are no more available points. MI has the longest exploration path and the lowest exploration efficiency due to its one-sided exploration strategy of pursuing high-gain regions. LS also exhibits a high proportion of exploration due to the inclusion of the FR strategy in its exploration strategy, which continuously directs it to the nearest frontier point for exploration. The GS has significant performance degradation in the later stages of exploration, but has good exploration efficiency in the smaller-sized test maps 1 and 2. However, in the larger test map 3, exploration efficiency dropped significantly and is not as good as the LS.

The two experiments described above lead us to the following conclusions:LAGS has a stronger exploration performance. In cases with different initial positions, LAGS can solve the regional legacy issues and plan reasonable exploration paths during the exploration process. In addition, LAGS can achieve a higher exploration ratio for the same average path length compared to other algorithms.LAGS has good robustness and environmental adaptability. In test experiments in environments of various sizes and layouts, LAGS achieves better performance with shorter exploration paths and higher exploration efficiencies compared to other algorithms.

## 7. Conclusions

In this paper, we analyze the impact of the regional legacy issues prevalent in existing exploration algorithms on exploration efficiency and propose a LAGS algorithm to address the problem that a single exploration strategy is challenging to balance between optimal exploration paths and environmental robustness. The algorithm addresses the redundant path and region legacy issues by integrating a local exploration strategy with a global perception strategy to improve exploration efficiency. Our experiments, conducted in environments with various layouts and sizes, demonstrate the robustness, environmental adaptability, and high exploration efficiency of the LAGS algorithm. The algorithm allows the exploration of unknown environments faster, to speed up the search for victims in missions such as personnel search and rescue and disaster reconnaissance. In addition, we use GPR and BO sampling points as candidate action points for the robot, which provides a new method for the robot to select candidate action points.

In future research, we will consider speeding up the training process of the A3C network by introducing empirical information gained during human deployment into the algorithm, e.g., by applying inverse reinforcement learning techniques. Secondly, we aim to optimize the exploration strategy further to reduce the system’s decision-making time. Finally, autonomous exploration in unknown environments with dynamic obstacles is also one of our possible subsequent research directions.

## Figures and Tables

**Figure 1 sensors-23-04766-f001:**
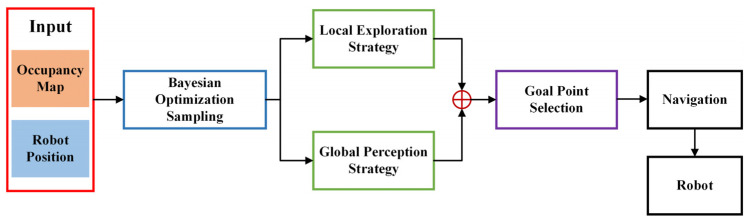
Proposed system pipeline diagram.

**Figure 2 sensors-23-04766-f002:**
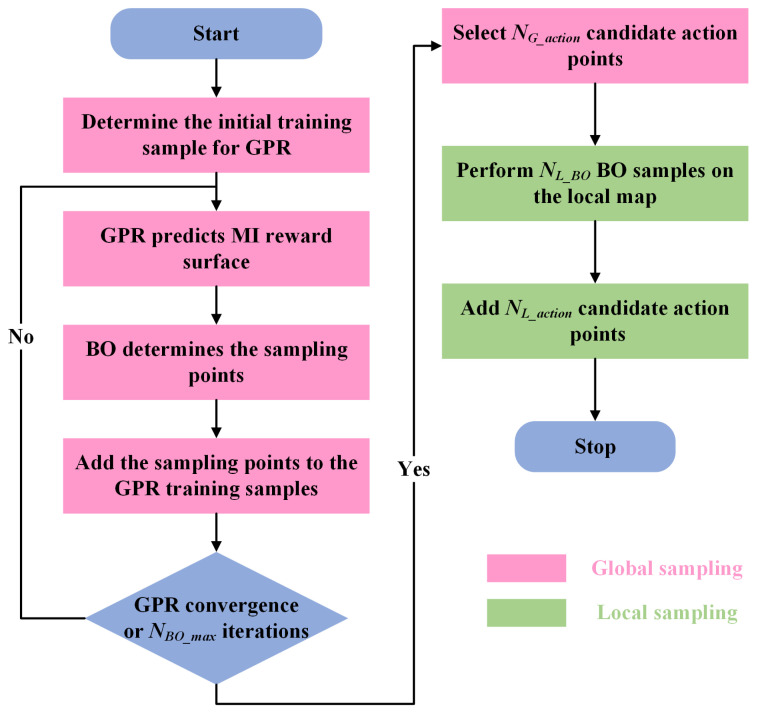
BO sampling flow diagram.

**Figure 3 sensors-23-04766-f003:**
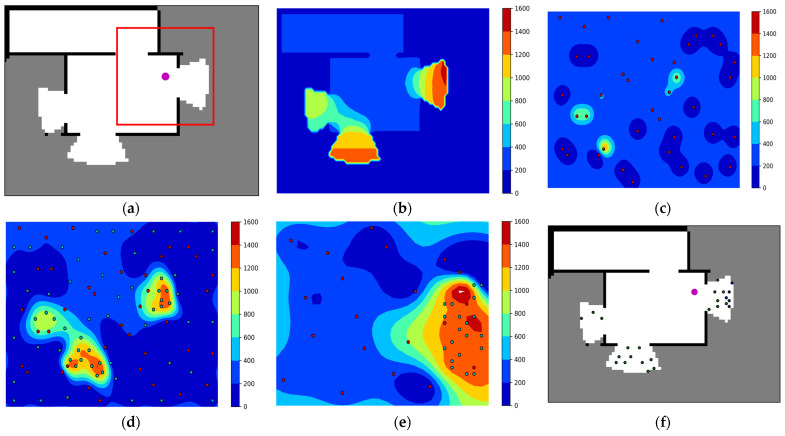
Bayesian optimal sampling process. (**a**) Current occupancy map, robot position, and local map; (**b**) Ground Truth: Real MI reward surface; (**c**) MI reward surface and initial training samples; (**d**) MI reward surface and training samples after 60 BO iterations; (**e**) Local MI reward surfaces and local sampling points; (**f**) Selected candidate action points (global: red, local: blue).

**Figure 4 sensors-23-04766-f004:**
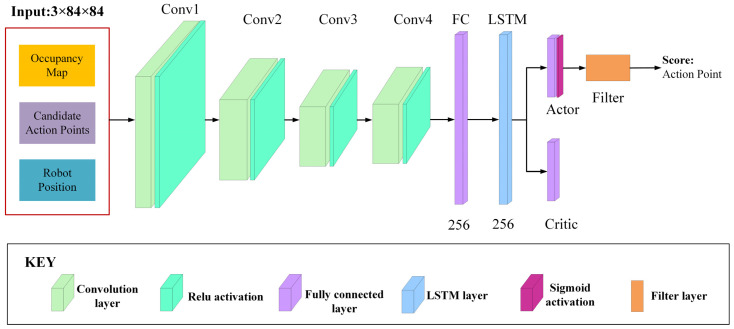
System network structure.

**Figure 5 sensors-23-04766-f005:**
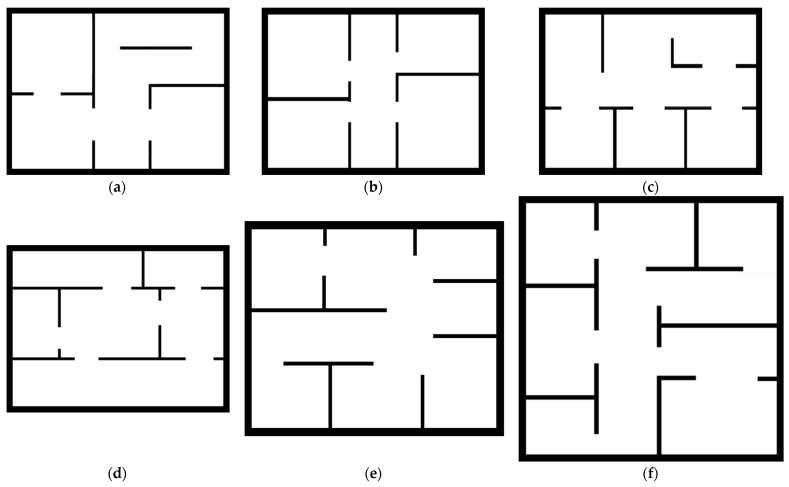
Training and test maps. (**a**) Train map 1; (**b**) Train map 2; (**c**) Train map 3; (**d**) Test map 1; (**e**) Test map 2; (**f**) Test map 3.

**Figure 6 sensors-23-04766-f006:**
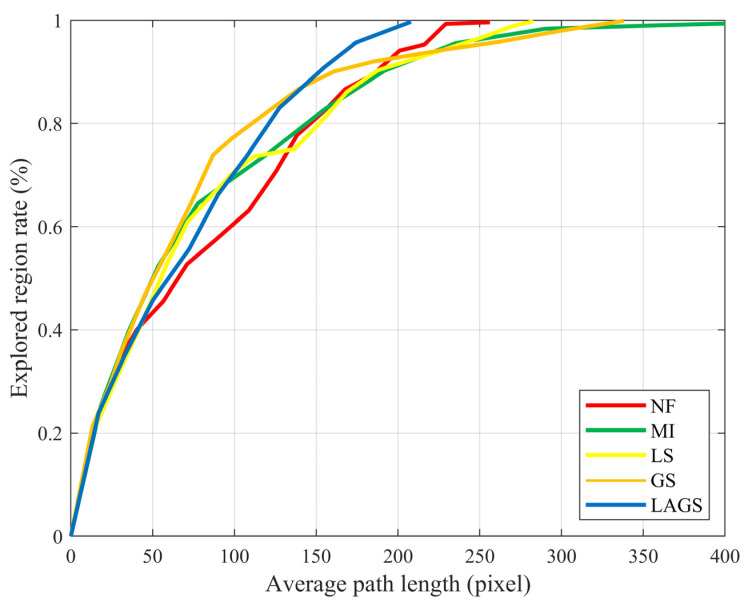
Proportion of exploration at average path length.

**Figure 7 sensors-23-04766-f007:**
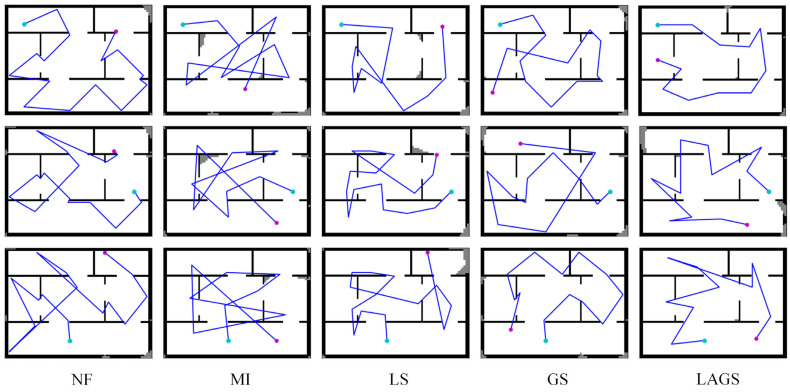
Exploration paths for each algorithm at initial positions (10,10) (top), (70,35) (middle), and (35,50) (bottom), respectively. The cyan point is the initial position of the exploration, the magenta point is the end position of the exploration, and the blue line is the line between two adjacent exploration target points and is not the actual path of movement of the robot.

**Figure 8 sensors-23-04766-f008:**
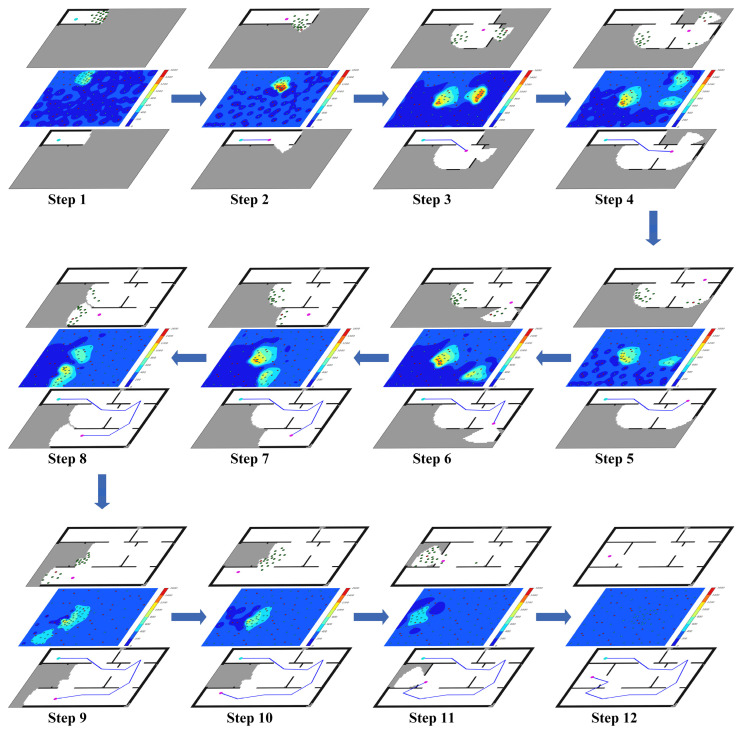
The exploration process of the LAGS algorithm at an initial position of (10,10). The entire process consists of 12 steps, each with three layers; the lower layer includes the current occupancy map, the robot’s initial position (cyan), the robot’s current position (magenta), and the line between previously adjacent target points (blue), the middle layer includes the MI reward surface, and sampling points predicted based on the current occupancy map; and the upper layer includes the robot’s current position (magenta), multiple candidate action points (green), and candidate target points (red).

**Figure 9 sensors-23-04766-f009:**
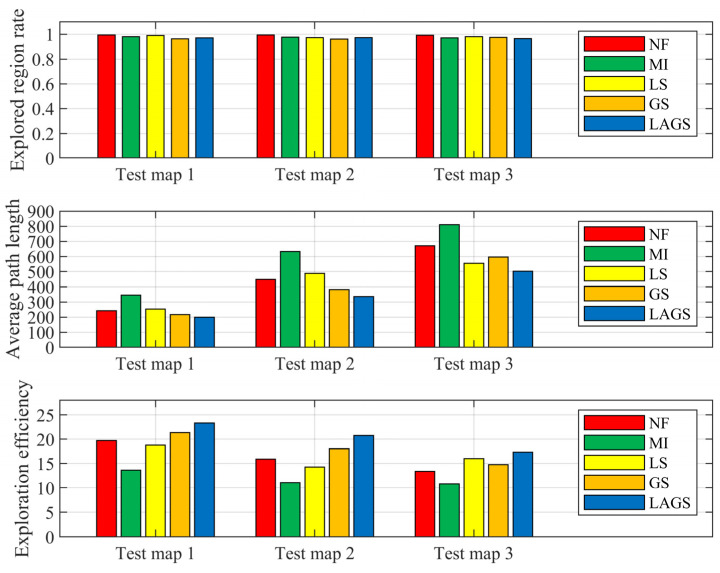
Statistical results of the different algorithms in the test maps.

**Table 1 sensors-23-04766-t001:** Details of training and test maps.

Map Name	Resolution (Pixel)
Train map 1	60 × 80
Train map 2	60 × 80
Train map 3	60 × 80
Test map 1	60 × 80
Test map 2	77 × 93
Test map 3	95 × 95

**Table 2 sensors-23-04766-t002:** A3C network hyperparameters.

Hyperparameters	Value
Number of parallel environments	3
Number of minibatches T	30
Number of episodes	100,000
Learning rate	0.0001
Learning rate decay policy	Polynomial decay
Optimization algorithm	Adam
Value loss coefficient α	0.5
Entropy coefficient β	0.01
Discount factor γ	0.99
Exploration region rate ρ	0.95
Score proportion factor η	0.75
Covariance smoothness coefficient v	2.5
BO tradeoff coefficient κ	5.0

**Table 3 sensors-23-04766-t003:** Network structure hyperparameters.

Layer	Hyperparameters
Conv1	Output = 16, Kernel = 8, Stride = 4, Padding = Valid
Conv2	Output = 32, Kernel = 4, Stride = 2, Padding = Valid
Conv3	Output = 32, Kernel = 3, Stride = 2, Padding = Same
Conv4	Output = 32, Kernel = 3, Stride = 2, Padding = Same
FC	Output size = 256
LSTM	Output size = 256

**Table 4 sensors-23-04766-t004:** Comparison of statistical results between different algorithms.

	Average Path Length			Explored Region Rate			Exploration Efficiency		
	Test Map 1	Test Map 2	Test Map 3	Test Map 1	Test Map 2	Test Map 3	Test Map 1	Test Map 2	Test Map 3
NF	242.19	448.4	670.79	0.994	0.995	0.993	19.7	15.89	13.36
MI	344.86	633.07	809.82	0.98	0.976	0.97	13.64	11.04	10.81
LS	253.69	488.45	554.03	0.991	0.972	0.981	18.75	14.25	15.98
GS	216.61	381.47	595.76	0.963	0.961	0.975	21.34	18.04	14.77
LAGS	199.57	335.79	501.96	0.97	0.973	0.965	23.33	20.75	17.35

## Data Availability

The raw/processed data required to reproduce these findings cannot be shared at this time as the data also forms part of an ongoing study.

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
