# Peer review of "Efficient Autonomous Exploration and Mapping in Unknown Environments"

_sensors, 2023, doi:10.3390/s23104766_

Round 1
Reviewer 1 Report
How about the completeness criteria of the proposed algorithm. Is it able to find a path if one exists?
Is the algorithm capable of finding path through a small passage?
Is it able to be executed in real-time?
Author Response
Thank you very much for reviewing our manuscript entitled "Efficient Autonomous Exploration and Mapping in Unknown Environments" and giving very valuable comments. I will answer all your questions in detail below.
Comment 1: How about the completeness criteria of the proposed algorithm. Is it able to find a path if one exists?
Response to Comment 1:
Thank you for your valuable comment.
Our proposed LAGS algorithm will select the target point for the robot's next movement based on the current state (including the current occupancy map, robot position, and candidate action points) and then drive the robot to move to the target point for exploration.
In terms of candidate action point selection, we calculate action points with high gain based on the mutual information surface of the currently occupied map as candidate action points for the robot's next move, so if there are unexplored regions in space, the algorithm will always select action points near unexplored regions as candidate action points for exploration.
The robot moves to the target point using the more mature A* algorithm, so when there is a feasible path in space, the robot can plan the path and move to the target point for exploration.
Therefore, if there are feasible paths and unexplored areas in space, the LAGS algorithm will keep driving the robot to explore until the current environment has been explored.
Comment 2: Is the algorithm capable of finding path through a small passage?
Response to Comment 2:
Thank you for your valuable comment. The selection of candidate action points for the robot relies on the mutual information surface of the currently occupied map. As long as a small channel contains an unexplored region, then it has a certain amount of information gain on the mutual information surface, and the LAGS algorithm will select a candidate action point in that vicinity for exploration. Thus, the LAGS algorithm also has good exploration performance in small channel regions.
Comment 3: Is it able to be executed in real-time?
Response to Comment 3:
Thank you for your valuable comment. The longest operation of the whole algorithm is the calculation of the mutual information surface of the currently occupied map. We use GPR and BO to predict the mutual information surface, which greatly reduces the running time of the algorithm and allows the algorithm to run in real-time on higher-performance embedded platforms (e.g., NVIDIA Jetson AGX Xavier), and there is still a lot of space for improvement, which is one of the directions of our future algorithm optimization.
Reviewer 2 Report
This paper presents the LAGS algorithm, which addresses regional legacy issues and improves exploration efficiency by combining local and global exploration strategies in autonomous robots. Experimental results demonstrate that the LAGS algorithm exhibits superior robustness, environmental adaptability, and exploration efficiency compared to other heuristic-based and learning-based methods. There are a few areas in the paper that could be improved. Here are some specific suggestions:
1. The flow of the introduction is generally logical, but there are a few areas where the discussion seems to jump abruptly from one topic to another. For example, the mention of deep reinforcement learning (DRL) after discussing frontier-based and information-based methods could be better integrated. Consider adding a sentence or two to provide a smoother transition between these topics.
2. Some sentences could be rephrased for better clarity. For example, the sentence "In LAGS, the local exploration strategy focuses on high-gain regions in the local map to explore unknown regions faster." could be rephrased as "In LAGS, the local exploration strategy emphasizes high-gain regions in the local map to expedite exploration of unknown areas.",
3. You've done a good job of identifying the limitations of existing methods in the field, including the regional legacy issues and the challenges associated with DRL-based exploration methods. However, you could more explicitly state how your proposed method, LAGS, addresses these research gaps.
4.The motivation for the research is clear, as the paper focuses on improving exploration efficiency and map quality in autonomous robots. Consider emphasizing the practical implications of these improvements, such as how they could lead to better performance in real-world applications like search and rescue or disaster reconnaissance.
5. The introduction provides an extensive overview of the existing literature, covering both heuristic-based and learning-based exploration methods. However, the organization of this section could be improved to make it easier to follow. Consider breaking this discussion into separate paragraphs, with one paragraph focused on heuristic-based methods and another on learning-based methods.
6. The narrative of the literature is comprehensive, but the introduction would benefit from a more explicit summary of the state of the field. Consider adding a sentence or two that synthesizes the main findings or trends in the existing literature.
7. The contributions of the paper are clearly enumerated in a numbered list, which is helpful for the reader. However, the text could be more concise in describing these contributions.
8. The grammar and phrasing of the text could be improved in several places. For example, consider changing "extended time can be seen during each decision process" to "each decision process can take an extended amount of time".
9. In related Works section some sentences could be rephrased for better clarity. For example, consider changing "Researchers have proposed various improvements based on this, for example, using a rapidly-exploring random tree (RRT) algorithm to find candidate frontier points" to "Researchers have proposed various improvements to this method, such as using a rapidly-exploring random tree (RRT) algorithm to identify candidate frontier points."
10. In related Works section Ensure a smooth flow within each paragraph by using transitions and topic sentences. For instance, the paragraph on learning-based methods starts by mentioning the potential of deep learning and reinforcement learning but then abruptly discusses DRL. Consider adding a sentence that connects these ideas more directly.
11. In related Works section Maintain consistency in the use of abbreviations and terminology throughout the section. For example, "DRL" is used in the introduction, but "DL" and "RL" are used separately in this section. Ensure that abbreviations are consistently applied and properly introduced.
12. At the end of the section, explicitly state how your proposed method, LAGS, builds upon or addresses the limitations of the methods discussed in the related works. This will help the reader understand the significance and novelty of your research.
13. There are a few instances where the grammar and phrasing of the text could be improved. For example, consider changing "Among all the effective works, we note that few researchers consider the regional legacy issues during exploration." to "Despite the numerous effective works in the field, we observe that few researchers address the regional legacy issues during exploration." Additionally, ensure proper punctuation, such as using a colon instead of a hyphen in "The development of autonomous exploration can generally be divided into two categories:".
14. Discuss potential limitations and challenges faced by the LAGS algorithm.
Identify any limitations or challenges encountered during the experiments and provide insights into how they could be addressed in future work.
15. Conclusion
- Some sentences could be rephrased for better clarity. For example, consider changing "The algorithm solves the problems of redundant paths and the regional legacy issues that arise during exploration by combining a local exploration strategy with a global perception strategy." to "The LAGS algorithm addresses redundant paths and regional legacy issues by integrating a local exploration strategy with a global perception strategy."
- Consider revising "Experiments conducted in environments of various layouts and sizes show that the LAGS algorithm has good robustness and environmental adaptability and has high exploration efficiency." to "Our experiments, conducted in environments with various layouts and sizes, demonstrate the robustness, environmental adaptability, and high exploration efficiency of the LAGS algorithm."
- Change "In future research, we will continue optimizing the exploration strategy to reduce the system's decision-making time." to "In future research, we aim to further optimize the exploration strategy to reduce the system's decision-making time."
- Briefly reiterate how your research addresses the limitations or gaps identified in the related works section, showcasing the novelty and relevance of your work.
- Consider adding a statement on the potential real-world impact of your research, such as how the LAGS algorithm could benefit rescue robots, sweeping robots, or disaster reconnaissance robots mentioned in the introduction.
above
Author Response
Thank you very much for reviewing our manuscript entitled "Efficient Autonomous Exploration and Mapping in Unknown Environments" and giving very valuable comments. During the past few days, we have considered these comments carefully. The removed parts from the original paper are shown in red, and the changes are in blue.
Comment 1: The flow of the introduction is generally logical, but there are a few areas where the discussion seems to jump abruptly from one topic to another. For example, the mention of deep reinforcement learning (DRL) after discussing frontier-based and information-based methods could be better integrated. Consider adding a sentence or two to provide a smoother transition between these topics.
Response to Comment 1: Thank you for your valuable suggestions on our manuscript. We have modified it according to your instructions (pages 1 and 2).
Comment 2: Some sentences could be rephrased for better clarity. For example, the sentence "In LAGS, the local exploration strategy focuses on high-gain regions in the local map to explore unknown regions faster." could be rephrased as "In LAGS, the local exploration strategy emphasizes high-gain regions in the local map to expedite exploration of unknown areas."
Response to Comment 2: Thank you for your valuable suggestions to help us improve our manuscript. We have made revisions based on your instructions (page 2).
Comment 3: You've done a good job of identifying the limitations of existing methods in the field, including the regional legacy issues and the challenges associated with DRL-based exploration methods. However, you could more explicitly state how your proposed method, LAGS, addresses these research gaps.
Response to Comment 3: Thanks to your valuable suggestions, we have added the relevant descriptions in this revision (page 2).
Comment 4: The motivation for the research is clear, as the paper focuses on improving exploration efficiency and map quality in autonomous robots. Consider emphasizing the practical implications of these improvements, such as how they could lead to better performance in real-world applications like search and rescue or disaster reconnaissance.
Response to Comment 4: Thanks to your valuable suggestions, we have added the relevant descriptions in this revision (page 2).
Comment 5: The introduction provides an extensive overview of the existing literature, covering both heuristic-based and learning-based exploration methods. However, the organization of this section could be improved to make it easier to follow. Consider breaking this discussion into separate paragraphs, with one paragraph focused on heuristic-based methods and another on learning-based methods.
Response to Comment 5: Thank you for your valuable suggestions on our manuscript. We consider the introduction to be only a cursory introduction to the two categories of exploration methods intended to capture the reader's interest in reading them, and we have provided a specific introduction to the two categories of exploration methods in related work.
Comment 6: The narrative of the literature is comprehensive, but the introduction would benefit from a more explicit summary of the state of the field. Consider adding a sentence or two that synthesizes the main findings or trends in the existing literature.
Response to Comment 6: Thanks to your valuable suggestions, we have added the relevant descriptions in this revision (page 2).
Comment 7: The contributions of the paper are clearly enumerated in a numbered list, which is helpful for the reader. However, the text could be more concise in describing these contributions.
Response to Comment 7: Thank you for your valuable suggestions on our manuscript. We have edited the section on points of innovation and technical contributions in this revision (page 3).
Comment 8: The grammar and phrasing of the text could be improved in several places. For example, consider changing "extended time can be seen during each decision process" to "each decision process can take an extended amount of time".
Response to Comment 8: Thank you for your valuable suggestions to help us improve our manuscript. We have made revisions based on your instructions (page 1 and 2).
Comment 9: In related Works section some sentences could be rephrased for better clarity. For example, consider changing "Researchers have proposed various improvements based on this, for example, using a rapidly-exploring random tree (RRT) algorithm to find candidate frontier points" to "Researchers have proposed various improvements to this method, such as using a rapidly-exploring random tree (RRT) algorithm to identify candidate frontier points."
Response to Comment 9: Thank you for helping us to improve our manuscript by improving the wording. We have made revisions based on your instructions (page 3).
Comment 10: In related Works section Ensure a smooth flow within each paragraph by using transitions and topic sentences. For instance, the paragraph on learning-based methods starts by mentioning the potential of deep learning and reinforcement learning but then abruptly discusses DRL. Consider adding a sentence that connects these ideas more directly.
Response to Comment 10: Thank you for your valuable suggestions on our manuscript. We have modified it according to your instructions (pages 4).
Comment 11: In related Works section Maintain consistency in the use of abbreviations and terminology throughout the section. For example, "DRL" is used in the introduction, but "DL" and "RL" are used separately in this section. Ensure that abbreviations are consistently applied and properly introduced.
Response to Comment 11: Thank you for your valuable suggestions on our manuscript. We have modified it according to your instructions (pages 4).
Comment 12: At the end of the section, explicitly state how your proposed method, LAGS, builds upon or addresses the limitations of the methods discussed in the related works. This will help the reader understand the significance and novelty of your research.
Response to Comment 12: Thank you for your valuable suggestions for our manuscript. We have already covered this in some detail in the introduction section, and repeated presentations would cause discomfort to readers.
Comment 13: There are a few instances where the grammar and phrasing of the text could be improved. For example, consider changing "Among all the effective works, we note that few researchers consider the regional legacy issues during exploration." to "Despite the numerous effective works in the field, we observe that few researchers address the regional legacy issues during exploration." Additionally, ensure proper punctuation, such as using a colon instead of a hyphen in "The development of autonomous exploration can generally be divided into two categories:".
Response to Comment 13: Thank you for your review of our manuscript. After having corrected the mistakes you pointed out, we double-checked the revised manuscript to ensure no other mistakes in our paper.
Comment 14: Discuss potential limitations and challenges faced by the LAGS algorithm.
Identify any limitations or challenges encountered during the experiments and provide insights into how they could be addressed in future work.
Response to Comment 14: Thank you for your valuable suggestions to help us improve our manuscripts. In this revision we have added a corresponding description in the summary section (page 19).
Comment 15.1: Some sentences could be rephrased for better clarity. For example, consider changing "The algorithm solves the problems of redundant paths and the regional legacy issues that arise during exploration by combining a local exploration strategy with a global perception strategy." to "The LAGS algorithm addresses redundant paths and regional legacy issues by integrating a local exploration strategy with a global perception strategy."
Response to Comment 15.1: Thank you for your guidance on the grammar of our manuscript. We have amended it in accordance with your instructions (page 19).
Comment 15.2: Consider revising "Experiments conducted in environments of various layouts and sizes show that the LAGS algorithm has good robustness and environmental adaptability and has high exploration efficiency." to "Our experiments, conducted in environments with various layouts and sizes, demonstrate the robustness, environmental adaptability, and high exploration efficiency of the LAGS algorithm."
Response to Comment 15.2: Thank you for your valuable suggestions. We have amended it in accordance with your instructions (page 19).
Comment 15.3: Change "In future research, we will continue optimizing the exploration strategy to reduce the system's decision-making time." to "In future research, we aim to further optimize the exploration strategy to reduce the system's decision-making time."
Response to Comment 15.3: Thank you for your guidance on the wording of our manuscript. We have revised it in accordance with your instructions (page 19).
Comment 15.4: Briefly reiterate how your research addresses the limitations or gaps identified in the related works section, showcasing the novelty and relevance of your work.
Response to Comment 15.4: Thank you for your kind comments on our manuscript. In this revision we have added a corresponding description in the summary section (page 19).
Comment 15.5: Consider adding a statement on the potential real-world impact of your research, such as how the LAGS algorithm could benefit rescue robots, sweeping robots, or disaster reconnaissance robots mentioned in the introduction.
Response to Comment 15.5: Thank you for your valuable suggestions to help us improve our manuscripts. In this revision we have added a corresponding description in the summary section (page 19).
Reviewer 3 Report
The paper addresses an important problem related to autonomous systems and methods of perception of the robot's environment. In my opinion, the presented algorithm, although it indeed draws on previous scientific results, it is quite original.
However, I have some critical comments with respect to the manuscript.
1. In section 4, the authors characterise the optimisation method used. They give specific parameter values, for example the number of sampling points, the number of iterations, etc. However, there is a lack of discussion as to why the values of these parameters were chosen in this way. It seems that it would have been better to use some symbols to describe the parameters rather than their values. This gives a more general impression of the method. A similar comment can also be made with regard to section 5 and especially 5.1.2. In contrast, the network parameters summarised in Tables 2 and 3 are better presented, although the strategy for selecting these parameters and the network structure is not explained.
2. I believe that the paper lacks a description of even an elementary model of the robot's movement. It is not clear whether we are talking about an omnidirectional robot, a nonholonomic robot with differential drive or a car. It should be noted that a non-holonomic constraint may make the task of exploration more difficult due to the inability to move in any direction.
3. It is not clear why the paths shown in Fig. 7 intesect walls in the environment. How should these results be interpreted?
4. There is no analysis of the influence of the algorithm parameters and the structure of the neural network on the results obtained. Are the algorithm parameters used critical? Were they chosen based on a number of examples or are they the result of the authors' intuition? I think these questions are quite important, especially in the context of the neural networks used. In such cases, one can get the impression that the structure of the network is the result of chance rather than tests to find even a suboptimal solution.
The presentation style of the mathematical formulas is not well chosen. In particular, references to equations before they are defined should be avoided - there is no need to use numbers just before the mathematical formulas are defined - cf. (5), (6), (7), (8), (9), (12), (15), (16), (17), (18), (19), (20),
Author Response
Thank you very much for reviewing our manuscript entitled "Efficient Autonomous Exploration and Mapping in Unknown Environments" and giving very valuable comments. During the past few days, we have considered these comments carefully. The removed parts from the original paper are shown in red, and the changes are in blue.
Comment 1: In section 4, the authors characterise the optimisation method used. They give specific parameter values, for example the number of sampling points, the number of iterations, etc. However, there is a lack of discussion as to why the values of these parameters were chosen in this way. It seems that it would have been better to use some symbols to describe the parameters rather than their values. This gives a more general impression of the method. A similar comment can also be made with regard to section 5 and especially 5.1.2. In contrast, the network parameters summarised in Tables 2 and 3 are better presented, although the strategy for selecting these parameters and the network structure is not explained.
Response to Comment 1: Thank you for your valuable comments to help us improve the manuscript. The number of points sampled, the number of iterations, and the number of candidate action point in section 4 and the maximum reward limit and hyperparameters for terminal actions in section 5 are empirical value derived from a number of tests. In this revision we use notation in section 4 to describe these parameters and add an explanation of how the parameters were chosen.
Comment 2: I believe that the paper lacks a description of even an elementary model of the robot's movement. It is not clear whether we are talking about an omnidirectional robot, a nonholonomic robot with differential drive or a car. It should be noted that a non-holonomic constraint may make the task of exploration more difficult due to the inability to move in any direction.
Response to Comment 2: Thank you for your valuable suggestions. The LAGS algorithm drives the robot exploration by publishing the target point for the next movement. The robot moves to the target point using more established path planning algorithms such as A*, Dijkstra. So, the LAGS algorithm can usually be applied to any chassis structure, including but not limited to omnidirectional, differential, or Ackermann structures. In the experimental part of this paper, we assume that the robot is omnidirectional and add a corresponding description to the simulation description in section 6.2.
Comment 3: It is not clear why the paths shown in Fig. 7 intesect walls in the environment. How should these results be interpreted?
Response to Comment 3: Thank you for your critical comment, and we apologize for the misunderstanding caused by the lack of clarity in our description. As the actual path of the robot is too cluttered to be displayed in the diagram, we have used a more concise line between the target points to describe the general path of the robot's movement. We have added a corresponding description in the figure caption to figure 7 to avoid misunderstandings on the part of the reader.
Comment 4: There is no analysis of the influence of the algorithm parameters and the structure of the neural network on the results obtained. Are the algorithm parameters used critical? Were they chosen based on a number of examples or are they the result of the authors' intuition? I think these questions are quite important, especially in the context of the neural networks used. In such cases, one can get the impression that the structure of the network is the result of chance rather than tests to find even a suboptimal solution.
Response to Comment 4: Thank you for your valuable suggestions. The neural network structure used in this paper was inspired by the following code and papers:
- https://github.com/shareeff/A3C_CNN_LSTM
- https://github.com/MatheusMRFM/A3C-LSTM-with-Tensorflow
- https://github.com/xushsh163/A3CSuperMario_Windows
- Niroui, K. Zhang, Z. Kashino and G. Nejat, "Deep Reinforcement Learning Robot for Search and Rescue Applications: Exploration in Unknown Cluttered Environments," in IEEE Robotics and Automation Letters, vol. 4, no. 2, pp. 610-617, April 2019, doi: 10.1109/LRA.2019.2891991.
The neural network structure in codes (a), (b), and (c) is basically the same, and the network structure has been widely used in games such as SuperMario, Doom, and Atari with good results, so in this paper, we borrow the network structure and add two convolutional layers for better extraction of image features. At the Filter layer of the network, we have borrowed from the network design in (d) so that the network can be better adapted for use in autonomous exploration tasks.
For the hyperparameter settings, we have referred to the hyperparameters in the code in (a), (b), and (c) and have conducted several experiments using different values for important hyperparameters such as number of minibatches, number of episodes, and learning rate in order to improve the performance of the network in autonomous exploration tasks. We have added a description of the hyperparameter settings in the 6.3 Training Setup.
Comment 5: The presentation style of the mathematical formulas is not well chosen. In particular, references to equations before they are defined should be avoided - there is no need to use numbers just before the mathematical formulas are defined - cf. (5), (6), (7), (8), (9), (12), (15), (16), (17), (18), (19), (20).
Response to Comment 5: Thank you for reviewing the formulas in our manuscript. After correcting the errors you pointed out, we double-checked the formulas in the revised manuscript to ensure no other errors in the formulas in our paper.
Reviewer 4 Report
The work presents a path-planning technique combining local and global information to select the appropriate robot's path. The manuscript is well-written, with clear results and comprehensive experiments. Their finding deserves publication. No comments from the referee.
Author Response
Thank you very much for reviewing our manuscript entitled "Efficient Autonomous Exploration and Mapping in Unknown Environments" and for your kind comments on our manuscript.
Round 2
Reviewer 2 Report
The authors revised the manuscript according to the comments and the paper is suitable for publication in its current form
none